# EvoPlan: Agent-driven Evolutionary Planning for LLM Reasoning

## Abstract

Efficiently generating high-quality plans is a critical yet unsolved challenge for Large Language Model (LLM) agents tackling complex reasoning tasks. Prevailing search-based planners, such as those employing Monte Carlo Tree Search (MCTS) or exploring a tree-of-thoughts, are fundamentally bottlenecked by their reliance on costly, execution-based rollouts to evaluate partial solutions, leading to prohibitive computational overhead. We introduce a novel agentic planning framework that circumvents this limitation by replacing expensive execution with efficient, static evaluation. Our framework employs a duo of specialized LLM critics: a Logical Consistency Agent to scrutinize a plan's internal coherence and a Feasibility Agent to assess its practical executability. These critics provide rich, multi-faceted feedback that guides a novel evolutionary search algorithm, which iteratively refines complete candidate plans toward global optimality. On diverse mathematical reasoning benchmarks (e.g., GSM8K, AIME), our approach surpasses vanilla MCTS by +8.72pp while using 90% less GPU time, and outperforms LLM-based search by +7.66pp with 30% fewer search steps. Our work demonstrates that decoupling plan evaluation from execution through specialized agentic critics enables a more scalable and effective framework for LLM-based planning and reasoning.

## 1 Introduction

Enabling Large Language Models (LLMs) to solve complex, multi-step problems has driven a shift from simple auto-regressive generation (Wei et al., 2022) toward deliberate, search-based planning (Wei et al., 2025; Li, 2025). The current state of the art is dominated by step-level search methods, which explore a tree of intermediate thoughts. These methods are fundamentally bottlenecked by their evaluation mechanism: assessing a partial plan's value requires either noisy LLM-generated heuristics (Yao et al., 2023; Kambhampati et al., 2024) or costly, execution-based rollouts (Hao et al., 2024), limiting the search depth and scope.

A more powerful, yet hitherto infeasible, strategy is to reframe planning as a global optimization problem over *complete* candidate plans. Evolutionary algorithms offer a natural paradigm for this global search (Guo et al., 2024; Yang et al., 2024; Wei et al., 2025; Li, 2025), but their application has been blocked by the prohibitive cost of fitness evaluation. This "evaluation bottleneck (Zhang et al., 2024b; Wang et al., 2024b; Kambhampati et al., 2024) stems from the same core issue: assessing a plan's quality has historically required full, costly execution, making it impossible to evolve a large population of plans within a reasonable computational budget.

We introduce **EvoPlan**, a novel agentic planning framework that makes this global optimization strategy viable. It achieves this by framing LLM planning as an **evolutionary search over complete plans, guided by an efficient, execution-free fitness function.** The core of our approach is a duo of specialized LLM critics that serve as static evaluators: a *Logical Consistency Agent* to scrutinize a plan's internal coherence and a *Feasibility Agent* to assess its practical executability. This agentic fitness function circumvents the execution bottleneck, allowing EvoPlan to run a tree-based evolutionary search that iteratively refines complete plans toward a global optimum, without performing a single execution during the search phase. Our experiments show this leads to substantial gains: on GSM8K, EvoPlan surpasses a state-of-the-art MCTS-based planning baseline (RAP (Hao et al., 2024)) by **+8.72pp** in accuracy while using **90% less GPU time**.

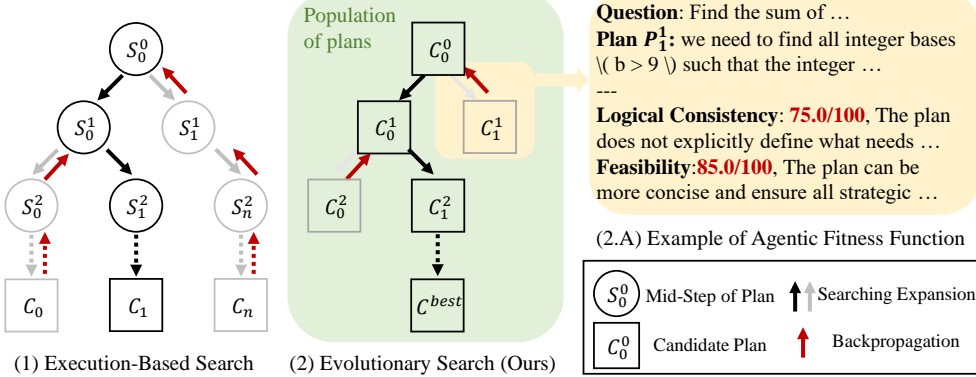

Figure 1: **EvoPlan solves the evaluation trap by replacing costly execution with lightweight agentic critique.** (1) **Execution-Based Search** methods like MCTS (Hao et al., 2024) search a tree of intermediate states (*circles*, $S_i$). To evaluate a single state, they must perform expensive execution-based rollouts, completing the plan and running it (*squares*, $C_n$). This creates an evaluation trap: the prohibitive cost limits search depth, leading to many suboptimal branches (gray arrow and nodes). (2) **EvoPlan's Evolutionary Search** breaks this trap by optimizing over a population (marked in *green*) of complete candidate plans (*squares*, $C_i$). The core innovation is our efficient, execution-free **agentic fitness function** (2.A). A duo of critic agents provides each plan with a quantitative fitness score (e.g., Logic: 75.0, Feasibility: 85.0) and qualitative feedback. This rich signal is backpropagated (*red arrow*) to guide the evolutionary search toward a globally optimal plan ($C^{\text{best}}$). By decoupling evaluation from execution, EvoPlan enables scalable global optimization.

Figure 1 provides a conceptual overview of the EvoPlan framework. It highlights the decoupled, agent-driven loop where a Planner agent proposes mutations to a population of plans, a duo of parallel Critic agents provide a rich fitness score, and a search controller directs the evolution. The single, most promising plan is then passed to a dedicated Executor. Our contributions are:

1. **A novel agentic planning framework, EvoPlan,** that reframes LLM reasoning as an evolutionary search over complete plans to enable global optimization.
2. **An efficient, execution-free agentic fitness function** composed of specialized LLM critics, which enables scalable evolutionary search by replacing costly execution-based evaluation with lightweight static analysis.
3. **A novel adaptation of Monte Carlo Tree Search (MCTS)** as a plan-level evolutionary engine, where each node represents a complete plan and the reward signal is derived directly from our agentic fitness scores.
4. **An empirical finding that plan refinement and plan execution are separable reasoning skills** with different capability requirements. This insight, enabled by our framework's design, provides a blueprint for more efficient, hybrid-model agent systems.

## 2 EVOPLAN: AGENT-DRIVEN EVOLUTIONARY PLANNING

EvoPlan reframes LLM reasoning as an evolutionary search process (Zhang et al., 2024b; Li, 2025), designed to discover globally optimal plans. Its core innovation is an agentic evaluation mechanism that breaks the computational bottleneck of traditional, execution-based search methods (e.g., Yao et al., 2023; Hao et al., 2024), as shown in Table 1. This section first formalizes our general framework for evolutionary planning and then describes its specific instantiation using a novel adaptation of MCTS.

### 2.1 A FRAMEWORK FOR EVOLUTIONARY PLANNING VIA AGENTIC JUDGMENT

The objective of EvoPlan is to find an optimal plan $C^*$ from a space of candidates $\mathcal{C}$ that maximizes a quality function $Q(C)$. Formally, we seek:

$$C^* = \arg\max_{C \in \mathcal{C}} Q(C). \tag{1}$$

Table 1: **Positioning EvoPlan in the landscape of LLM planning frameworks.** Our framework decouples evaluation from execution by using specialized *agentic critics*, breaking the cost-accuracy trade-off that constrains prior approaches.

| Aspect | EvoPlan (Ours) | Execution-Based Search | Heuristic-Based Search | Auto-regressive Gen. |
|---|---|---|---|---|
| Representative works | (this work) | RAP (Hao et al., 2024) MR (Gao et al., 2024) | ToT Yao et al. (2023) | Plan-and-Solve (Wang et al., 2023) |
| Search Algorithm | Evolutionary (MCTS) | MCTS | BFS/DFS | None (Greedy) |
| **Primary Evaluation Signal** | **Agentic Critics** | **Full Plan Execution** | **LLM Heuristics / Votes** | **None** |
| Search Granularity | Complete Plans | Partial States | Partial States | N/A |
| Introduces Correction | Before Execution | During Search | During Search | No |
| Computational Cost[†] | $1\times$ | $9.17\times$ | $1.42\times$ | - |

[†]Normalized search cost relative to EvoPlan (lower is better).

We structure this search around three core evolutionary components, each implemented by a dedicated LLM agent:

- **Population and Individuals.** The population is a set of complete candidate plans. Each plan, a sequence of steps $C = (s_1, \ldots, s_L)$, constitutes an individual in the search space.

- **Mutation Operator.** A Planner Agent ($M_p$) serves as the mutation operator. It takes a parent plan $C$ and textual feedback $F$ to generate a set of $K$ refined offspring plans: $\{C'_k\}_{k=1}^K = M_p(C, F)$. This recasts the LLM as a structured proposal generator, a concept explored in recent work on LLM-driven optimization (Yang et al., 2024; Guo et al., 2024). Mutations include rewriting steps, altering granularity, or addressing specific flaws identified by critics.

- **Agentic Fitness Function.** Our central innovation is an efficient, execution-free fitness function composed of specialized LLM critics. While the use of LLMs for feedback or verification is an emerging area (Gero et al., 2023; Lyu et al., 2023; Zhang et al., 2024a), our approach is distinct in decomposing this judgment into specialized, efficient critics to form a static fitness function for evolutionary search. The critics are:

  1. **Logical Consistency Agent ($E_{LC}$):** Scrutinizes the plan for internal contradictions, returning a soundness score $S_{LC}(C) \in [0, 1]$.
  2. **Feasibility Agent ($E_F$):** Assesses whether each step is actionable and coherent, returning a practicality score $S_F(C) \in [0, 1]$.

The aggregate fitness score $R(C)$, which serves as the reward signal for the search, is a weighted combination of these scores:

$$R(C) = w_{LC}S_{LC}(C) + w_F S_F(C), \tag{2}$$

where $w_{LC} + w_F = 1$. The textual outputs of these critics provide the feedback $F$ for the mutation operator.

A naive evolutionary approach, such as a simple genetic algorithm, would fail to preserve the relational structure between parent and offspring plans. To efficiently guide the search, it is critical to track this evolutionary lineage, which motivates a structured, tree-based search algorithm.

## 2.2 TREE-BASED EVOLUTION VIA MCTS INSTANTIATION

We instantiate the EvoPlan framework with a tree-based evolutionary search, leveraging MCTS for its principled approach to balancing exploration and exploitation. This approach has proven effective in domains ranging from classic games (Silver et al., 2016) to LLM reasoning (Hao et al., 2024). In our formulation, we adapt the four standard MCTS phases (Chaslot et al., 2008) to operate on complete plans, guided by our agentic fitness function. The full pseudocode is provided in Section 2.2.

**Evolutionary Tree Representation.** We define a search tree where each node represents a complete plan (an individual). An edge from a parent node to a child node signifies a mutation event. Thus, the path from the root to any node represents that plan's direct evolutionary lineage. This structure allows the algorithm to credit or discredit entire branches of the evolutionary history, systematically focusing the search on promising regions of the plan space.

---

**Algorithm 1** EvoPlan Framework

---

**Require:** Problem $P$, initial plan $C_0$, LLM-based planner $M$, evaluators set $E$, LLM executor $L$
**Ensure:** Optimal plan $C^*$, Solution $Y$
 1: $root \leftarrow \text{Node}(C_0, 0, 0)$          ▷ Initialize root node with initial plan, Q=0, N=0
 2: $leaf\_queue \leftarrow [root]$          ▷ Initialize leaf queue with the root node
 3: **while** $leaf\_queue$ is not empty **do**
 4:  $path \leftarrow \text{SelectPromissingPath}(root)$     ▷ Get the path from root to current leaf
 5:  $leaf \leftarrow \text{Dequeue}(leaf\_queue, path)$    ▷ Get the next leaf node from the queue by path
 6:  **if** not IsTerminal($leaf$) **then**  ▷ Terminate if the plan is optimal (all eval agents give full score) or reaches the max depth
 7:   Expand($leaf$)          ▷ Add child nodes to current leaf
 8:   **for** each $child$ in $leaf.children$ **do**
 9:    $scores, feedback \leftarrow \text{Evaluate}(child, P, E)$  ▷ Use LLM Agnet evaluator to review current plan
10:    Backpropagate($path, scores$)     ▷ Update Q and N along the path
11:    Enqueue($leaf\_queue, child$)    ▷ Add the new child nodes to the queue
12:   **end for**
13:  **end if**
14: **end while**        ▷ Stop seaching if all plans in search tree has been fully explored
15: $C^* \leftarrow \text{SelectBestPlan}(root)$       ▷ Select the plan with highest value
16: $Y \leftarrow \text{ExecutePlan}(P, C^*, L)$
17: **return** $C^*, Y$

18: **function** EXPAND($node$)
19:  $modified\_plan \leftarrow \text{ModifyPlan}(node.plan, node.feedback, M)$  ▷ Use LLM to modify plan based on the feedback
20:  $new\_node \leftarrow \text{Node}(modified\_plan, node.problem)$   ▷ Create a new node based on modified plan
21:  $node[children] \leftarrow \{new\_node, new\_node, new\_node\}$  ▷ Add the new nodes as children of current node
22: **end function**

23: **function** EVALUATE($plan, P, E$)
24:  $scores \leftarrow \{\}$       ▷ Initialize an empty dictionary to store evaluation scores
25:  $feedback \leftarrow []$       ▷ Initialize an empty list to store feedback text
26:  **for** each evaluator in $E$ **do**
27:   $score, feedback \leftarrow evaluator.evaluate(P, C)$  ▷ Get evaluation score and text, as detailed in Section A.4
28:   $scores[evaluator.\text{class\_name}] \leftarrow score$    ▷ Store the score with evaluator name
29:   $feedback.append(feedback)$     ▷ Store the feedback text
30:  **end for**
31:  **return** $scores, feedback$      ▷ Return all scores and feedback
32: **end function**

---

**Selection.** Starting from the root node (an initial plan $C_0$), the algorithm traverses the tree by recursively selecting the child plan $C$ that maximizes the Upper Confidence Bound 1 (UCB1) score (Auer, 2000):

$$\text{UCB1}(C) = \bar{R}(C) + c_{\exp}\sqrt{\frac{\ln N_p}{N_C}}. \tag{3}$$

Here, $\bar{R}(C)$ is the plan's average fitness score from the critic agents (exploitation), $N_C$ is its visit count, $N_p$ is its parent's visit count, and $c_{\exp}$ is an exploration hyperparameter. This phase identifies the most promising evolutionary path to refine.

**Expansion and Mutation.** The selection process continues until a leaf node $C_L$ (a plan not yet mutated) is reached. The **Planner Agent** $M_p$ is then invoked to perform targeted mutation, analogous to the proposal step in LLM-based optimizers (Yang et al., 2024), generating $K$ offspring plans. These new plans become children of $C_L$ in the tree, expanding the search frontier.

**Evaluation (Fitness Calculation).** Each newly generated child plan $C'_k$ is evaluated by our duo of **Critic Agents**, $E_{LC}$ and $E_F$. This step provides the fitness score $R(C'_k)$ for the new individual. This agentic evaluation is the key to EvoPlan's efficiency. Unlike state-level MCTS methods like RAP (Hao et al., 2024), which require a costly, execution-based rollout for every evaluation, or heuristic-based approaches that use noisy LLM votes to guide the search (Yao et al., 2023), our method uses a small number of parallelizable inference passes over the plan text. This single step accounts for the order-of-magnitude reduction in computational cost shown in Table 1.

Table 2: **Accuracy (%) on GSM8K.** In a zero-shot setting, EvoPlan's global plan optimization outperforms state-level search and MCTS methods evaluated under similar conditions. Results from original publications are in *italics*.

| Planning Type | Method | Task Guidance | Search Space | Granularity | Model | GSM8K |
|---|---|---|---|---|---|---|
| Sequential Planning | **Plan-and-Solve** Kojima et al. (2022) | Zero-Shot | No search | Plan-once | *GPT-3* | *56.40* |
| | **Least-to-Most** Zhou et al. (2023) | Task Prompt | No search | Plan-once | *GPT-3* | *62.39* |
| LLM-based Planning | **Arrange & Execute** Qiu et al. (2024) | Task Fine-Tuned | Available Thoughts | State-Level | *Qwen2-7B-it* | *82.11* |
| | **Meta Reasoning** Gao et al. (2024) | Task Prompt | Available Functions | State-Level | *GPT-4* Qwen2.5-7B-it | *92.10* 91.02 |
| | **Tree-of-Thought** Yao et al. (2023) | Task Prompt | Available Thoughts | State-Level | *GPT-4* Qwen2.5-7B-it | *90.00* 89.95 |
| MCTS-based Planning | **RAP** Hao et al. (2024) | 4-Shots | Available Steps | State-Level | *Llama-33B* Qwen2.5-7B-it | *48.80* 83.09 |
| | **EvoPlan (Ours)** | Zero-Shot | Available Plans | Plan-Level | Qwen2.5-7B-it | **91.81** |

**Backpropagation.** The fitness score $R(C_k')$ for each new child is propagated up its lineage (the path to the root). The visit counts ($N_C$) and average fitness estimates ($\bar{R}(C)$) of all ancestor plans along this path are updated. This process reinforces successful evolutionary trajectories and informs subsequent selection decisions.

After a predefined search budget, the plan $C^*$ with the highest visit count is selected as the final, optimized plan. This plan is then passed to a separate **Executor Agent** for high-fidelity execution. This principled decoupling of search and execution, enabled by our agentic fitness function, is fundamental to EvoPlan's design and anticipates our findings on the benefits of heterogeneous agent roles (Zhong et al., 2024; Yao et al., 2025b).

## 3 EXPERIMENTS

Our experiments are designed to validate EvoPlan by making a three-part argument. We first establish that our agentic framework successfully circumvents the critical evaluation bottleneck that constrains prior search methods. We then demonstrate how this efficiency translates into superior plan quality and task performance. Finally, we leverage EvoPlan's decoupled architecture as a scientific instrument to reveal a key insight about the separability of reasoning skills in LLMs. Details on our experimental setup, including baselines, models, benchmarks, and hyperparameters, are provided in Appendix A.3.

### 3.1 SOLVING THE EVALUATION BOTTLENECK WITH AN AGENTIC FITNESS FUNCTION

A core claim of our work is that specialized agentic critics can form a valid and highly efficient substitute for costly, execution-based rollouts. To validate this, we first show that our critic agents provide a meaningful fitness signal, and second, that this mechanism is orders of magnitude more efficient than its execution-based counterpart.

**The Agentic Fitness Signal is Effective and Non-Redundant.** A useful fitness function must provide an informative signal to guide the evolutionary search. We test this through an ablation study on Qwen2.5-7B-it, with results in Table 4. Removing either the Logical Consistency ($E_{LC}$) or Feasibility ($E_F$) agent from the fitness function leads to a significant performance drop, with mean accuracy decreasing by 4.09pp and 4.02pp, respectively. The effect is especially pronounced on complex problems like AIME-24, where accuracy falls by 6.66pp without $E_{LC}$ and 9.16pp without $E_F$. This provides strong evidence that both critics offer essential and non-redundant signals, effectively guiding the search towards high-quality plans.

**Agentic Evaluation is Dramatically More Efficient.** The primary motivation for our agentic fitness function is to address the prohibitive cost of execution-based rollouts found in standard MCTS methods like RAP (Hao et al., 2024). Table 3b confirms EvoPlan's success, showing a dramatic efficiency improvement on the GSM8K benchmark. Using Qwen2.5-7B-it, EvoPlan attains a +8.72pp accuracy improvement over RAP while reducing GPU time by 90% (0.71 vs. 7.11 hours). This efficiency is further compounded by a more focused search; by operating on complete plans, EvoPlan requires 30% fewer search steps on average than the state-level heuristic search in ToT (Yao

Table 3: **Comparative efficiency analysis of EvoPlan**. (a) EvoPlan's plan-level search requires 30% fewer steps on average than ToT's state-level search. (b) On GSM8K, EvoPlan achieves higher accuracy than Execution-Based MCTS (RAP) while using 90% less GPU time, demonstrating the efficiency of its agentic evaluation.

(a) Fewer Search Steps vs. State-Level Search.

| Method | Model | Steps # Avg. |
|---|---|---|
| State-Level Search (ToT) | Qwen2.5-7B-it | 78351.33 |
| | Llama3.1-8B-it | 78402.33 |
| **EvoPlan** | Qwen2.5-7B-it | **54932.17** |
| | Llama3.1-8B-it | **54849.33** |

(b) Reduced GPU Cost vs. Execution-Based MCTS.

| Method | Model | GSM8K | GPU Hour | Eff. Ratio |
|---|---|---|---|---|
| Execution-Based MCTS (RAP) | Qwen2.5-7B-it | 83.09 | 7.11 | 11.68 |
| | Llama3.1-8B-it | 75.06 | 6.47 | 11.60 |
| **EvoPlan** | Qwen2.5-7B-it | **91.81** | **0.71** | **129.30** |
| | Llama3.1-8B-it | **75.42** | **0.77** | **97.94** |

Table 4: **Ablation of evaluation agents using Qwen2.5-7B-it.** Performance (Accuracy %) drops significantly when either the Logical Consistency or Feasibility agent is removed, confirming their individual importance as fitness signals for the evolutionary search.

| Evaluator | AIME-24 | AIME-25 | Olympiad | AddSub | GSM8K | SingleEQ | Avg. |
|---|---|---|---|---|---|---|---|
| Feasibility Only | $21.67_{\pm0.41}$ | $1.67_{\pm0.13}$ | $24.70_{\pm0.43}$ | $89.81_{\pm0.30}$ | $86.24_{\pm0.34}$ | $94.24_{\pm0.23}$ | 53.05 |
| Logic-Consistency Only | $19.17_{\pm0.40}$ | $3.33_{\pm0.18}$ | $24.93_{\pm0.43}$ | $89.62_{\pm0.31}$ | $87.36_{\pm0.33}$ | $94.34_{\pm0.23}$ | 53.12 |
| **Combined (EvoPlan)** | $\mathbf{28.33}_{\pm0.45}$ | $\mathbf{6.67}_{\pm0.25}$ | $\mathbf{29.81}_{\pm0.46}$ | $\mathbf{90.89}_{\pm0.29}$ | $\mathbf{91.81}_{\pm0.27}$ | $\mathbf{95.32}_{\pm0.21}$ | **57.14** |

Table 5: **Generality on Commonsense and Arithmetic Reasoning (Accuracy %).** EvoPlan significantly outperforms a strong baseline across general reasoning tasks, confirming its broad applicability.

| Model | Method | CommonsenseQA | MultiArith | SVAMP | Avg. |
|---|---|---|---|---|---|
| Qwen2.5-7B-it | Chain-of-Thought (ZS-CoT) | 63.72 | 95.33 | 83.40 | 80.82 |
| | **EvoPlan (Ours)** | **79.20** | **98.67** | **92.90** | **90.26** |
| Llama3.1-8B-it | Chain-of-Thought (ZS-CoT) | 63.80 | 38.17 | 27.00 | 42.99 |
| | **EvoPlan (Ours)** | **68.57** | **92.76** | **81.20** | **80.84** |
| **Avg. Improvement** | | **+10.12** | **+28.97** | **+31.85** | **+23.65** |

et al., 2023), as detailed in Table 3a. These results confirm we have developed an effective and computationally feasible fitness function, successfully solving the evaluation bottleneck that has hindered global plan optimization.

## 3.2 UNLOCKING GLOBAL OPTIMIZATION FOR SUPERIOR PERFORMANCE

Having established an efficient search mechanism, we now investigate whether this enables superior global optimization, leading to higher accuracy compared to state-level methods. On the widely-used GSM8K benchmark (Table 2), EvoPlan achieves 91.81% accuracy with Qwen2.5-7B-it in a zero-shot setting. This outperforms state-level planners like ToT (89.95%) and execution-based MCTS like RAP (83.09%), demonstrating the tangible benefit of our approach on a standard reasoning task.

To further probe the generality of our framework, we evaluated EvoPlan on standard commonsense and arithmetic reasoning benchmarks: CommonsenseQA (Talmor et al., 2019), MultiArith (Roy & Roth, 2015), and SVAMP (Patel et al., 2021). These tasks provide a complementary evaluation to the complex mathematical challenges. The results, presented in Table 5, show that EvoPlan consistently and significantly outperforms the zero-shot Chain-of-Thought baseline (Wei et al., 2022). On average, EvoPlan improves accuracy by +10.12pp on CommonsenseQA, +28.97pp on MultiArith, and +31.85pp on SVAMP. This demonstrates that the benefits of our evolutionary search, guided by an execution-free agentic fitness function, are not confined to a single domain. The framework's ability to globally optimize complete plans is a general principle that yields substantial performance gains across a variety of reasoning tasks.

Table 7: **Performance and Efficiency of Hybrid Model Configurations with EvoPlan.** Using smaller models for planning/evaluation (1.5B) and a large model for execution (72B) significantly reduces GPU hours while preserving 93.6% of the accuracy, demonstrating the separability of reasoning skills.

| Planner | Evaluator | Executor | AIME-24 | AIME-25 | Olympiad | AddSub | GSM8K | SingleEQ | Avg. | GPU Hour | Eff. Ratio |
|---|---|---|---|---|---|---|---|---|---|---|---|
| 1.5B | 72B | 72B | 20.00 | 6.67 | 31.41 | 90.38 | 90.07 | 97.83 | 56.06 | 14.39 | 3.89 |
| 1.5B | 72B | 1.5B | 13.33 | 0.00 | 13.48 | 70.13 | 61.33 | 76.57 | 39.14 | 10.41 | 3.75 |
| **1.5B** | **1.5B** | **72B** | **26.67** | **6.67** | **30.52** | **88.61** | **90.30** | **97.05** | **56.63** | **5.72** | **9.90** |
| 72B | 72B | 72B | 30.00 | 13.33 | 36.11 | 91.58 | 94.58 | 97.34 | 60.49 | 15.46 | 3.91 |

Table 6: **Accuracy (%) of EvoPlan vs. Baselines across diverse mathematical benchmarks.** EvoPlan demonstrates superior average performance, with significant gains on complex tasks, underscoring the benefit of its global, plan-level search. Results are mean accuracy with standard errors; highest values are in **bold**.

| Model | Method | AIME-24 | AIME-25 | Olympiad | AddSub | GSM8K | SingleEQ | Avg. |
|---|---|---|---|---|---|---|---|---|
| Llama3.1-8B-it | Plan and Solve | 13.02 $_{\pm 0.34}$ | **0.42** $_{\pm 0.06}$ | 12.87 $_{\pm 0.33}$ | 85.87 $_{\pm 0.35}$ | 72.74 $_{\pm 0.45}$ | 91.67 $_{\pm 0.28}$ | 46.10 |
| | Meta Reasoning | 13.33 $_{\pm 0.34}$ | 0.00 $_{\pm 0.00}$ | 13.93 $_{\pm 0.35}$ | 86.96 $_{\pm 0.34}$ | 72.78 $_{\pm 0.45}$ | 89.17 $_{\pm 0.31}$ | 46.03 |
| | Tree-of-Thought | **15.83** $_{\pm 0.37}$ | 0.00 $_{\pm 0.00}$ | 14.04 $_{\pm 0.35}$ | 83.80 $_{\pm 0.37}$ | 66.24 $_{\pm 0.47}$ | 84.30 $_{\pm 0.36}$ | 44.04 |
| | **EvoPlan (Ours)** | **15.83** $_{\pm 0.37}$ | 0.00 $_{\pm 0.00}$ | **19.07** $_{\pm 0.39}$ | **87.78** $_{\pm 0.33}$ | **75.42** $_{\pm 0.43}$ | **92.77** $_{\pm 0.26}$ | **48.48** |
| Qwen2.5-7B-it | Plan and Solve | 18.75 $_{\pm 0.39}$ | 4.58 $_{\pm 0.21}$ | 26.03 $_{\pm 0.44}$ | 87.89 $_{\pm 0.33}$ | 90.30 $_{\pm 0.30}$ | 94.05 $_{\pm 0.24}$ | 53.60 |
| | Meta Reasoning | 13.33 $_{\pm 0.34}$ | 0.00 $_{\pm 0.00}$ | 19.41 $_{\pm 0.40}$ | 88.48 $_{\pm 0.32}$ | 90.98 $_{\pm 0.29}$ | 93.11 $_{\pm 0.25}$ | 50.89 |
| | Tree-of-Thought | 20.00 $_{\pm 0.40}$ | 3.33 $_{\pm 0.18}$ | 28.89 $_{\pm 0.45}$ | 87.59 $_{\pm 0.32}$ | 90.62 $_{\pm 0.29}$ | 93.95 $_{\pm 0.24}$ | 54.06 |
| | **EvoPlan (Ours)** | **28.33** $_{\pm 0.45}$ | **6.67** $_{\pm 0.25}$ | **29.81** $_{\pm 0.46}$ | **90.89** $_{\pm 0.29}$ | **91.81** $_{\pm 0.27}$ | **95.32** $_{\pm 0.21}$ | **57.14** |

### 3.3 An Empirical Finding: The Separability of Reasoning Skills

Finally, we test whether the cognitive skill of iterative *plan refinement* is fundamentally different from, and less complex than, the skill of high-fidelity *plan execution*. EvoPlan's modular architecture allows us to probe this question by deploying models of vastly different scales to each role. We assign small, efficient models (`Qwen2.5 1.5B`) to the iterative planning and criticism phases, reserving a large, powerful model (`Qwen2.5 72B`) only for the final, one-shot execution of the optimized plan.

Table 7 presents the results. The hybrid 1.5B/72B system achieves an average accuracy of 56.63%. This is a striking result, as it retains 93.6% of the performance of a much more costly system where the 72B model is used for all stages (60.49% accuracy). This high performance is achieved with a 63% reduction in GPU hours (from 15.46 to 5.72 hours), yielding a 2.5x improvement in the overall efficiency ratio (pass@1 accuracy per GPU hour).

This is more than a simple efficiency gain; it is a key empirical finding. The ability of a small 1.5B model to effectively guide the search for a powerful 72B model provides strong evidence that LLM reasoning is not monolithic. The skills required to compare, critique, and incrementally refine plans appear separable from, and less computationally demanding than, the skills for flawless execution. This insight offers a principled path for designing more sophisticated, resource-aware, and scalable multi-agent reasoning systems.

## 4 Discussion

Our experimental results validate EvoPlan as a state-of-the-art reasoning framework and, more importantly, utilize its architecture to uncover a fundamental insight into LLM reasoning. By reframing planning as an evolutionary process guided by an agentic fitness function, we achieve significant gains in both accuracy and efficiency. Here, we discuss the primary scientific contribution enabled by our work, its broader implications for designing agentic systems.

### 4.1 The Primary Insight: A Dichotomy of Reasoning Skills

A key scientific contribution of this work is the first empirical demonstration of a clear dichotomy between the cognitive skills of **plan refinement** and **plan execution**. Current paradigms for LLM reasoning largely treat these as a monolithic process, often deploying a single, powerful model for all sub-tasks. Our work provides strong evidence challenging this assumption.

This separability is most evident in our hybrid model experiment (Table 7). A system using a small 1.5B model for the iterative refinement phase (planning and criticism) and a powerful 72B model for the final, one-shot execution retained 93.6% of the accuracy of an end-to-end 72B system. This result is not merely an efficiency gain; it is a scientific finding. It reveals that the abilities required for iterative comparison, critique, and refinement of plans are computationally less demanding than the ability for flawless, high-fidelity execution. This dynamic can be conceptualized as a "manager" versus "expert" relationship: the planner and critic agents act as managers that assess strategy and direction, a task requiring strong comparative judgment rather than exhaustive domain knowledge. In contrast, the executor acts as an expert that carries out the final, approved strategy, a task requiring deep and precise knowledge.

## 4.2 Implications for Agentic AI System Design

The demonstrated dichotomy provides a new, empirically grounded blueprint for designing the next generation of multi-agent AI systems. Current approaches that rely on teams of homogeneous, powerful agents are not only computationally inefficient but also conceptually underdeveloped. Our findings suggest a more principled path toward agent specialization.

First, this work validates the design of **heterogeneous agent teams**, where roles are assigned based on the cognitive complexity of the task at hand. Instead of deploying multiple expensive, generalist agents, systems can achieve a superior cost-performance trade-off by composing teams of smaller, specialized "manager" agents to deliberate over strategy, reserving the most powerful "expert" agents for final execution. Second, our execution-free, agentic fitness function enables what we term **scalable deliberation**. By decoupling the cost of evaluating a plan from its execution, an agent system can explore a vastly larger space of potential strategies within a given computational budget. This allows for more thorough and robust problem-solving, overcoming the prohibitive cost-per-thought that constrains existing search methods.

## 5 Related Work

LLM reasoning research has progressively incorporated search algorithms to overcome the myopic, greedy nature of auto-regressive generation. These methods, however, remain fundamentally constrained by a trade-off between evaluation cost and signal quality, a dilemma EvoPlan resolves through its unique agentic architecture.

**The Cost-Fidelity Dilemma in State-Level Search.** To move beyond single-path generation, methods like Tree-of-Thought (ToT) (Yao et al., 2023) and Meta-Reasoning (MR) (Gao et al., 2024) employ tree search to explore multiple intermediate reasoning steps. These frameworks operate at the **state-level** and fall into two distinct camps. On one end, heuristic-based search methods such as ToT and its generalization, Graph-of-Thoughts (Besta et al., 2024), rely on an LLM to provide a cheap but noisy evaluation signal, such as a "value" score or a "vote" on the most promising partial thought (Yao et al., 2023). While this enables broad exploration, the unreliable signal often leads to inefficient search. On the other end, execution-based search provides a high-fidelity signal at a prohibitive cost. For example, RAP (Hao et al., 2024) applies state-level MCTS, but its evaluation of each node requires a full, execution-based *rollout* using the LLM as a world model. This strong reward signal comes at the cost of extreme computational overhead. This cost-fidelity dilemma creates an "evaluation trap" that has prevented truly global optimization over the complete plan space.

**The Bottleneck in Evolutionary Plan Optimization.** A compelling alternative is to perform global optimization over complete plans, a paradigm well-suited for evolutionary algorithms from Genetic Programming (GP) Xu et al. (2024); Zhang et al. (2024b). This approach has been historically blocked by the "fitness evaluation bottleneck," where assessing a large population of candidates is computationally infeasible (Zhang et al., 2024b; Wang et al., 2024b; Surina et al., 2025). Recent works have adeptly used LLMs as components in evolutionary systems, for example to evolve prompts (e.g., EvoPrompt (Guo et al., 2024)), act as optimizers (e.g., OPRO (Yang et al., 2024) and its successors (Yuksekgonul et al., 2025; Liu et al., 2025; Xiang et al., 2025)), or design reward functions (e.g., RE-GoT (Yao et al., 2025a)). However, they do not resolve the core bottleneck, as they typically rely on a single, monolithic, and expensive LLM to judge fitness. EvoPlan directly

breaks this impasse. Its innovation is not just using agents for evaluation, but using an **agentic fitness function that is decomposed** into a duo of lightweight, specialized critics (Logical Consistency and Feasibility). This design provides the fast and reliable signal required to make evolutionary search a viable and powerful strategy for global LLM plan optimization.

**Architectural Niche in Agentic AI Systems.** EvoPlan's design also carves out a distinct niche within the landscape of multi-agent AI systems (OpenAI, 2024; Wang et al., 2024a; Wu et al., 2024). Instead of following the conversational structure of systems like AutoGen (Wu et al., 2024), or emulating human-centric software development workflows like MetaGPT (Hong et al., 2024) and ChatDev (Qian et al., 2024), EvoPlan's architecture is organized around a **computational search paradigm**: evolutionary MCTS. This structure enables a mechanism of procedural refinement that is distinct from the post-hoc, inter-trial learning of frameworks like Reflexion (Shinn et al., 2023). Rather than learning only after a complete, failed attempt, EvoPlan's Planner and Critic agents engage in a continuous, intra-trial refinement loop to optimize a plan *before* committing to a single execution. By instantiating this specialized structure, EvoPlan serves as strong empirical validation for the emerging principle of heterogeneous agent teams (Zhong et al., 2024; Yao et al., 2025b; Chen et al., 2025), providing a clear architectural blueprint that distinguishes it from both homogeneous agent systems and those that simply mirror human organizational patterns.

EvoPlan thus provides a unified solution. By introducing an execution-free agentic fitness function, it makes the previously infeasible strategy of global evolutionary search at the plan-level both practical and effective. In doing so, our work resolves the cost-fidelity dilemma of state-level search and provides a new, computationally-grounded architecture for multi-agent reasoning.

# 6 LIMITATIONS AND FUTURE HORIZONS

The principles and architecture of EvoPlan establish a foundation for several exciting avenues for future research, highlighting the generality and power of our approach.

First, while we demonstrated EvoPlan's efficacy in the structured domain of mathematical reasoning, its architecture serves as a robust testbed for planning in more open-ended domains like software engineering, scientific discovery, or business strategy. Adapting the critic agents to these new contexts is a clear and promising direction for extending the framework's impact.

Furthermore, the performance of EvoPlan is guided by the capabilities of its agentic critics. The modularity of these critics is a core strength of our framework, as it invites future work on enhancing their judgment mechanisms. For instance, one could integrate retrieval-augmented generation to ground their evaluations in external knowledge bases, thereby increasing their robustness and domain-specific expertise without altering the core evolutionary search algorithm.

Finally, we instantiated our evolutionary framework using a novel adaptation of MCTS, but the central principle of an agentic fitness function is algorithm-agnostic. This opens a rich research direction for exploring other evolutionary strategies, such as genetic algorithms or direct policy optimization, to guide the search over the plan population. EvoPlan thus serves as a general platform for investigating the intersection of evolutionary computation and agentic LLM reasoning.

# 7 CONCLUSION

We introduced **EvoPlan**, a framework that reframes LLM planning as an evolutionary search over complete plans. Its core mechanism is an execution-free, **agentic fitness function** that resolves the evaluation bottleneck of prior search methods. This innovation enables an MCTS-based evolutionary search that achieves state-of-the-art accuracy with order-of-magnitude efficiency gains. More importantly, our decoupled architecture serves as a scientific instrument, revealing a fundamental dichotomy between the capabilities required for plan refinement and plan execution. EvoPlan thus offers both a powerful method for robust reasoning and a principled blueprint for designing the next generation of scalable, agentic AI systems.

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

# A    APPENDIX

## A.1    ETHICS STATEMENT

This research was conducted in accordance with the ICLR Code of Ethics. The study did not involve human participants or animal subjects. All datasets are publicly available and were handled in strict compliance with their usage policies and licensing, ensuring no personally identifiable information was used. Our evaluation protocols were designed to be fair and impartial, centered on objective performance metrics.

## A.2    LLM USAGE

We utilized large language models as assistive tools for manuscript preparation, code debugging, and literature review. The use of all models complied with their respective terms of service.

## A.3    IMPLEMENTATION FOR REPRODUCIBILITY

This appendix provides supplementary details on our experimental setup, hyperparameters, and the specific prompt templates used to guide our specialized LLM agents.

**Reproducibility and Environment.**    To ensure reproducibility and facilitate fair comparisons, we have published our implementation and demonstrations in the supplementary material. All experiments were conducted within the official `lmsysorg/sglang` Docker container from the SGLang platform (Zheng et al., 2024), using 8 NVIDIA A100 GPUs.

**Code Availability.**    We attached the demo code as supplementary material for reproducibility.

**Evaluation Framework.**    Our evaluation framework extends the established benchmarking pipeline from CoT (Wei et al., 2022) and its recent refinements (Kong et al., 2024). For mathematical benchmarks, we required answers to be formatted within `\boxed{}` for consistent extraction. We used `Qwen2.5-7B-it` as a fallback extractor for any malformed outputs. Answer verification employed both exact matching and the `Math Verify` (HuggingFace, 2025) tool, following community best practices.

**Baselines.**    We integrated several baseline methods into our unified evaluation pipeline to enable comprehensive comparisons. These included Plan-and-Solve (Wang et al., 2023) and Tree-of-Thought (ToT) (Yao et al., 2023), which were adapted from their original implementations. We also implemented Meta Reasoning (Gao et al., 2024) following the descriptions in the official paper. For the GSM8K-specific comparison, we incorporated RAP (Hao et al., 2024) using the authors' official codebase. This unified approach ensures consistent assessment criteria across all methods.

**Datasets.**    We evaluated EvoPlan across a diverse set of benchmarks. For mathematical reasoning, we used AIME-24 and AIME-25 (MAA, 2024; 2025), Math Olympiad (He et al., 2024), AddSub (Hosseini et al., 2014), GSM8K (Cobbe et al., 2021), and SingleEQ (Koncel-Kedziorski et al., 2015). To test general reasoning skills, we included CommonsenseQA (Talmor et al., 2019), MultiArith (Roy & Roth, 2015), and SVAMP (Patel et al., 2021).

**Hyperparameters.**    For the EvoPlan framework, the MCTS-based evolutionary search was configured with the following key hyperparameters. The UCB1 exploration constant $c_{\exp}$ was set to $1.5$. In the expansion phase, the Planner Agent generated $K = 3$ offspring plans (mutations) for each selected leaf node. The total search budget was set to a maximum of 32 evaluation steps per problem. The agentic fitness score $R(C)$ was computed with equal weights for the critic agents ($w_{LC} = 0.5$, $w_F = 0.5$).

## A.4 AGENT PROMPT TEMPLATES

The EvoPlan framework is composed of several specialized agent roles, each guided by distinct system and user prompts. This design ensures reproducibility by providing precise instructions for each agent's task. The prompts for the Initial Plan Generator, the Planner (mutation operator), the Critic agents, and the final Executor are detailed below.

**Initial Plan Generator Prompt**    This prompt is used once at the beginning of the search to generate the initial plan ($C_0$), which serves as the root node of the MCTS tree. It instructs the LLM to create a high-level strategic plan without implementation details.

---

**Initial Plan Generator Prompt**

**User**:
# Problem
{problem}

# Your Task
Create a clear, strategic plan to solve the provided problem. The plan must provide high-level guidance without diving into implementation details.

# Plan Requirements
- **Strategic Steps**: Each step must be a high-level strategic action that guides toward the solution without specifying implementation details.
- **Clear Numbering**: Use proper numbering (1., 2., 3.) and sub-steps when needed (1.1, 1.2, etc.).
- **Logical Sequence**: Arrange steps in logical order where each step builds upon previous ones.
- **Appropriate Abstraction**: Keep steps general and abstract. Avoid specific technical details or exact procedures.
- **Concise but Complete**: Include all necessary strategic elements while avoiding redundancy.
- **Global Guidance**: Focus on what needs to be accomplished rather than how to accomplish it.
# Example Format
1. [First strategic objective]
2. [Second strategic objective]
2.1 [Strategic sub-objective if needed]
2.2 [Another strategic sub-objective if needed]
3. [Third strategic objective]

Your output should contain ONLY the numbered plan with no additional commentary or explanation. Keep it abstract and focused on strategic guidance.

---

**Planner Agent (Mutation) Prompt**   The Planner Agent acts as the mutation operator. It takes a plan and feedback from the critic agents to generate a refined offspring plan.

---

**Planner Agent (Mutation) Prompt**

**System Prompt**: You are an expert strategic planner who creates improved, high-level plans based on feedback.

**User Prompt**:
# Problem
{problem}

# Current Plan
{plan}

# Feedback to Address
{feedback}

# Your Task
Create a significantly improved strategic plan that addresses the identified weaknesses while maintaining appropriate abstraction and global guidance focus.

**Improvement Guidelines:**
1. **Address Feedback**: Carefully review and directly address each piece of feedback provided.
2. **Maintain Abstraction**: Keep steps at a strategic level. Avoid diving into implementation specifics or technical details.
3. **Improve Structure**: Ensure proper numbering (1., 2., 3.) and use sub-steps (1.1, 1.2) where appropriate.
4. **Optimize Flow**: Rearrange or modify steps to create a more logical strategic sequence.
5. **Eliminate Redundancy**: Remove unnecessary or duplicate steps while ensuring strategic completeness.
6. **Enhance Clarity**: Make strategic guidance clearer without adding unnecessary implementation detail.
7. **Global Focus**: Ensure the plan provides comprehensive strategic direction rather than step-by-step execution.
# Required Output Format
Provide your response as a structured object with two keys:
- `plan`: The full text of the new, improved strategic plan with proper numbering and high-level, abstract steps.
- `changes_made`: A detailed list of the specific structural and content changes you made to address the feedback while maintaining abstraction.

---

**Logical Consistency Agent Prompt**   The Logical Consistency Agent ($E_{LC}$) evaluates a plan's internal coherence, providing a key component of the agentic fitness score.

---

**Logical Consistency Agent Prompt**

**System Prompt**: You are an expert plan evaluator specializing in logical consistency and plan structure.

**User Prompt**:
# Problem Context
{question}

# Plan to Evaluate
{plan}

{history_section}
# Your Task
Your job is to identify logical flaws, structural issues, and missing steps in the plan. Focus on ensuring the plan is logically sound and well-structured at a strategic level.

**Evaluation Criteria:**
1. **Logical Flow**: Verify that steps follow a logical sequence where each step builds upon the previous ones and leads naturally to the next.
2. **Contradictions**: Identify any contradictions, impossible steps, or conflicting instructions within the plan.
3. **Completeness**: Check for any critical missing steps or logical gaps that would prevent successful problem resolution.
4. **Step Structure**: Ensure steps are properly numbered (1., 2., 3.) with sub-steps clearly indicated (1.1, 1.2, etc.) when needed.
5. **Abstraction Level**: Verify that the plan maintains appropriate abstraction without diving into implementation details.
6. **Feedback Adherence**: If history is provided, verify if the new plan has successfully addressed the previous logical concerns.
# Required Output Format
Provide your evaluation as a structured response with the following keys:
- `score`: A score from 0 to 100 based on the rubric (scores 0-60 indicate Unacceptable; 60-70 Major Issues; 70-80 Minor Issues; 80-100 Good).
- `feedback`: A brief, high-level explanation focusing on logical flow and structural soundness.
- `suggestions`: A list of specific, actionable suggestions to improve logical consistency and plan structure.

---

**Feasibility Agent Prompt**   The Feasibility Agent ($E_F$) evaluates a plan's strategic clarity and practicality, providing the other component of the fitness score.

---

**Feasibility Agent Prompt**

**System Prompt**: You are an expert plan evaluator specializing in feasibility, strategic clarity, and practical guidance.

**User Prompt**:
# Problem Context
{question}

# Plan to Evaluate
{plan}

{history_section}
# Your Task
Your job is to assess if the plan provides clear strategic guidance while maintaining appropriate abstraction. Focus on ensuring the plan offers comprehensive direction without diving into implementation details.

**Evaluation Criteria:**
1. **Strategic Clarity**: Each step must provide clear strategic direction that guides toward the solution (avoid implementation details or specific technical execution).
2. **Appropriate Abstraction**: Steps should offer high-level guidance rather than detailed instructions. They should be general enough to allow flexible execution.
3. **Comprehensiveness**: The plan should cover all major strategic aspects needed to solve the problem without being overly prescriptive.
4. **Clarity**: Instructions must be unambiguous at the strategic level while avoiding unnecessary detail.
5. **Conciseness**: The plan should be as brief as possible while remaining strategically complete.
6. **Feedback Adherence**: If history is provided, verify if the new plan has successfully addressed the previous strategic concerns.
# Required Output Format
Provide your evaluation as a structured response with the following keys:
- `score`: A score from 0 to 100 based on the rubric (scores 0-60 indicate Unacceptable; 60-70 Major Issues; 70-80 Minor Issues; 80-100 Good).
- `feedback`: A brief, high-level explanation focusing on strategic clarity and appropriate abstraction.
- `suggestions`: A list of specific suggestions to improve strategic guidance while maintaining abstraction.

---

**Executor Agent Prompt**   The Executor Agent is invoked once the search is complete. It takes the final, optimized plan ($C^*$) and executes it to produce the final answer.

---

**Executor Agent Prompt**

**System Prompt**: You are a helpful AI assistant.

**User Prompt**:
# Problem
{problem}

# Plan to Execute
{plan}

Let's execute the plan step-by-step to solve the problem.

---

A.5 CASE STUDY: EVOLUTIONARY PLAN REFINEMENT

To provide a concrete illustration of EvoPlan's dynamics, this section presents a step-by-step walk-through of the evolutionary search process for a single mathematical problem. We demonstrate how the framework iteratively refines an initial plan toward an optimal solution, guided by the agentic fitness function provided by the Logical Consistency and Feasibility critics.

**Problem Statement.** Find the sum of all integer bases $b > 9$ for which $17_b$ is a divisor of $97_b$.

**1. Initial Plan Generation (Root Node).** The process begins by generating an initial high-level plan. This plan serves as the root node of the MCTS tree.

> **Initial Plan (Node: cbd60065...)**
>
> To solve the problem, we need to find all integer bases $b > 9$ such that the integer represented by $17_b$ divides the integer represented by $97_b$, and then sum those bases.

This initial plan is then evaluated by the critic agents. The feedback highlights its lack of procedural detail, which is reflected in a modest score.

> **Critic Feedback and Scores**
>
> **Logical Consistency (Score: 75.0/100):** The plan has a good logical flow... However, it lacks the step-by-step process for converting the numbers from base $b$ to base 10 and checking the divisibility condition.
> **Feasibility (Score: 85.0/100):** The plan provides clear strategic direction... However, it can be slightly more concise and ensure all major strategic aspects are comprehensively covered.

**2. Iterative Refinement via MCTS.** The initial feedback guides the Planner Agent (mutation operator) to generate improved offspring plans. The MCTS algorithm selects promising nodes for expansion. After several iterations, the search converges on a significantly improved plan. For instance, the plan at node '41530cf8...' (depth 3) demonstrates clear evolutionary progress.

> **Final Optimized Plan (Node: 41530cf8...)**
>
> 1. Convert the base-$b$ numbers $17_b$ and $97_b$ to their decimal (base-10) equivalents using the formula: $a_n b^n + ... + a_0 b^0$. For $17_b$, this becomes $1 \times b + 7$. For $97_b$, this becomes $9 \times b + 7$.
> 2. Check the divisibility condition: Determine if the decimal equivalent of $97_b$ (i.e., $9b + 7$) is divisible by the decimal equivalent of $17_b$ (i.e., $b + 7$). This involves performing the division and verifying if the remainder is zero.
> 3. Identify all bases $b > 9$ that satisfy the divisibility condition by systematically testing values starting from $b = 10$.
> 4. Sum the valid bases that meet the criteria.
> 5. Provide the final answer as the sum of all valid bases.

This refined plan receives higher scores from the critics, particularly for its logical consistency, because it now explicitly outlines the required conversion and verification steps.

> **Critic Feedback and Scores**
>
> **Logical Consistency (Score: 88.0/100):** The plan is well-structured and logically sound, addressing the problem effectively. It covers all necessary steps and follows a clear sequence.
> **Feasibility (Score: 85.0/100):** The revised plan effectively addresses the feedback and provides clear strategic direction. It maintains an appropriate level of abstraction while covering all major aspects needed to solve the problem.

**3. Final Plan Selection and Execution.** The MCTS search explores a tree of candidate plans, using the agentic fitness scores to guide its exploration. Figure 2 visualizes the final search tree and presents the quantitative analysis that led to the selection of the best plan. The node '41530cf8...' was selected due to its highest average reward.

```
Top 5 Nodes by Average Reward:

1. Node: 41530cf8..., Visits: 1, Avg Reward: 0.865
2. Node: d54426eb..., Visits: 3, Avg Reward: 0.855
3. Node: 9f92b3c5..., Visits: 5, Avg Reward: 0.853
4. Node: 77446f49..., Visits: 1, Avg Reward: 0.850
5. Node: 0b290482..., Visits: 1, Avg Reward: 0.850

MCTS Tree Visualization (Root:  [cbd60065 D:0 V:3 Q:0.567])
+-- [cbd60065 D:0 V:3 Q:0.567] To solve the problem, we need...
    +-- [77446f49 D:1 V:1 Q:0.850] 1. Convert the numbers...
    +-- [9f92b3c5 D:1 V:5 Q:0.853] 1. Convert the base-b...
        +-- [0b290482 D:2 V:1 Q:0.850] 1. Convert the base-b...
        +-- [d54426eb D:2 V:3 Q:0.855] 1. Convert the base-b...
            +-- [0c81408c D:3 V:1 Q:0.850] 1. Convert the base-b...
            +-- [41530cf8 D:3 V:1 Q:0.865] 1. Convert the base-b...
```

Figure 2: **MCTS search tree and node analytics.** The visualization shows the evolutionary lineage of plans. Each entry denotes '[Node ID (truncated) D:depth V:visits Q:avg_reward]'. The search correctly identifies the plan at depth 3 (Node '41530cf8...') as having the highest reward, demonstrating effective exploitation of promising evolutionary paths.

Finally, the selected optimal plan is passed to the Executor Agent, which follows the steps to produce the final, correct solution. This completes the process, showing how execution-free evolutionary refinement leads to a high-quality plan that enables successful problem solving.

---

**Final Solution from Executor Agent**

To solve the problem, we start by converting the base-$b$ numbers... For $9b + 7$ to be divisible by $b + 7$, $-56$ must be divisible by $b + 7$. Therefore, $b + 7$ must be a divisor of 56. Since $b > 9$, we have $b + 7 > 16$. The relevant divisors of 56 greater than 16 are 28 and 56.
- If $b + 7 = 28$, then $b = 21$.
- If $b + 7 = 56$, then $b = 49$.
The valid bases are $b = 21$ and $b = 49$. The sum of these bases is $21 + 49 = 70$.

$$\boxed{70}$$

