# OpenReview forum: "EvoPlan: Agent-driven Evolutionary Planning for LLM Reasoning"
_ICLR.cc/2026/Conference — Submitted to ICLR 2026_

### Official Review · Reviewer_Uc83 · 2025-10-23

**Soundness:** 2
**Presentation:** 3
**Contribution:** 2
**Rating:** 2
**Confidence:** 3

**Summary:**

This paper introduces EvoPlan. It searches over full candidate plans and uses two light LLM critics to score each plan for “logical consistency” and “feasibility.” An MCTS-style loop picks which plans to mutate and keep, then a final executor model runs only the selected plan. The pitch is simple: avoid expensive rollout-based evaluation at every node and replace it with cheaper, static critics. The paper reports solid accuracy on math-style benchmarks with lower compute than execution-heavy search.

**Strengths:**

1. Clean decoupling story: evaluation is separated from execution via two critics, so the search never needs to roll out full solutions during exploration
2. Concrete algorithm: plan-level tree, UCB1 selection, terminal condition, and pseudocode are spelled out; prompts for planner, critics, and executor are given in the appendix
3. Efficiency evidence: 90% less GPU time than RAP on GSM8K and about 30% fewer steps than ToT’s state-level search.
4. Ablations suggest both critics matter; removing either hurts accuracy across tasks.
5. Reasonable systems insight: using small models for planning and a large executor retains most accuracy while cutting cost, hinting at separable skills.

**Weaknesses:**

1. Overclaiming on “reasoning”: this is not a new reasoning framework inside the model. It is a search-and-evaluate procedure driven by external critics. Calling it a reasoning framework stretches the claim.
2. Cost is still material: yes, critics are cheaper than full rollouts, but the method still issues many LLM calls per instance; feeling like just transferring complexity from rollouts to tokens. The paper reports step counts and GPU hours, but not token-normalized costs, wall time per instance, or per-problem call budgets.
3. Tree construction is under-specified: nodes are complete plans and edges are mutations, but depth is not grounded in any task semantics. How depth is chosen, how mutation size is set, and how often early stopping triggers are not justified.
4. What exactly is EvoPlan: the paper sometimes reads like an eval workflow because the central novelty is the critic-based fitness. The pipeline does produce a final plan, so it is more than “just evaluation,” but the positioning is muddy.
5. Fitness reliability: there is no quantitative analysis of how critic scores correlate with final correctness. The risk of “gaming” the critics remains.

Minor Comments:
- Some comparisons are not fully same-model and same-budget. That weakens the headline gaps. Plus no confidence intervals or significance testing around the main accuracy numbers.
- Beyond math verifiers, other tasks rely on the executor’s answer format. That increases dependence on the final model.
- Stopping rules and budgets: sensitivity to evaluation-step budgets and the distribution of reached depths are not shown.

**Questions:**

See weakness.

---

### Official Review · Reviewer_KT7G · 2025-10-28

**Soundness:** 2
**Presentation:** 3
**Contribution:** 2
**Rating:** 4
**Confidence:** 3

**Summary:**

This paper addresses the computational cost of search-based planning for LLM reasoning. The paper proposes EvoPlan, which applies MCTS to complete plans instead of partial reasoning states. The key difference from prior work is replacing execution-based evaluation (e.g., Tree-of-Thoughts) with two prompted LLM "critics" that score plans on logical consistency and feasibility.

On GSM8K with Qwen2.5-7B, EvoPlan achieves 91.81% vs 83.09% for RAP, claiming 90% less GPU time (though it's unclear if this includes all relevant costs; see below). Results extend to AIME and other math benchmarks. An interesting hybrid experiment shows that using small models (1.5B) for planning/critics and large models (72B) only for execution retains 93.6% of performance with 63% less compute.

The work fits into recent efforts combining LLMs with evolutionary algorithms (EvoPrompt, LMEA) and tree search methods (ToT, RAP, LATS). The main contribution is showing that prompted LLM critics can substitute for execution in plan-level MCTS on math problems.

**Strengths:**

- **Addresses a real bottleneck.** Replacing execution-based evaluation with lightweight critic LLMs is a sensible idea that achieves meaningful efficiency gains. The 90% GPU reduction vs RAP (if the comparison is fair) would be valuable for practitioners.
- **Strong separability finding.** The result in Section 3.3 (Table 7) is the most interesting contribution. Showing that small 1.5B models can effectively guide search for a 72B executor is a practically useful insight for building resource-aware agent systems. This hybrid architecture result deserves more emphasis as it has implications beyond this specific method.
- **Solid empirical results on math.** Consistent improvements across GSM8K, AIME, and commonsense reasoning tasks demonstrate the method works in practice on its target domain.
- **Proper ablations.** Table 4 shows both critics are necessary and provide non-redundant signals, validating the design choices.
- **Clear presentation.** Helpful visualizations (Figure 1, appendix case study) aid understanding.

**Weaknesses:**

- **Limited evaluation scope.** All experiments are on mathematical/arithmetic reasoning. Claims about "global optimization" and general agent design would be stronger with at least one non-math domain (code generation, strategic planning, etc).
- **Modest technical contribution.** The "agentic fitness function" consists of two prompted LLMs that provide scores. While effective, framing this as a fundamental innovation or a solution to the "evaluation trap" overstates the contribution. Algorithm 1 reveals the simplicity: the Evaluate function (lines 23-32) is just a for loop over critic LLM calls. The MCTS adaptation is also straightforward - the algorithm runs standard MCTS but evaluates complete plans instead of partial states.
- **Comparison fairness concerns.** The efficiency comparison (Table 3b) claims 90% GPU reduction vs. RAP, but it's unclear whether the initial plan generation cost is included. This is critical since generating a complete initial plan is likely more expensive than generating a single reasoning step. Additional differences in prompting strategy (zero-shot vs 4-shot) further complicate the comparison. Clarification is needed to validate this key claim.
- **Missing failure analysis.** When do critics give high scores to incorrect plans? Table 6 shows EvoPlan is only marginally better than baselines on several datasets, suggesting the critics aren't always reliable. Related work ("When is Tree Search Useful for LLM Planning?", ACL 2024) found that LLM discriminators need >90% accuracy for tree search to significantly outperform simpler methods, and current LLMs often have 50%+ error rates. Without analyzing critic accuracy and failure modes, the claim of "solving the evaluation bottleneck" is questionable.
- **Potential planning-execution mismatch.** Plans generated during the search phase may be too complex or convoluted for the executor to handle effectively. More importantly, effective plans often need to account for execution-specific failure modes that only become apparent when actually executing steps. By separating these phases completely, the planner may generate strategies that look good on paper but fail in practice. The paper provides no analysis of this potential limitation.
- **Incomplete related work coverage.**  The related work section misses several relevant papers on plan-level search and decomposed planning/execution architectures:
  - DisCIPL (COLM 2025) - Decomposes reasoning into planning and execution phases, where a Planner model generates task-specific inference programs executed by Follower models, using Sequential Monte Carlo (SMC) instead of MCTS for search
  - Language Agent Tree Search (LATS; ICML 2024) - Combines MCTS with reflection for iterative plan refinement, very similar conceptually
  - Policy-Guided Tree Search (PGTS; ICML 2025) - Uses learned policies to guide tree search over reasoning paths
  - LMEA (arXiv 2024) - LLM-driven evolutionary algorithms for combinatorial optimization
  - Self-Refine (NeurIPS 2023) - Iterative refinement with self-feedback, essentially the same idea as the critic agents
  - LLM-Modulo Framework (arXiv 2024) - Combines LLM generators with model-based critics for plan validation

**Questions:**

1. Table 3b reports 90% GPU time reduction vs RAP, which is a top-line claim of the paper. However, it's unclear whether this includes the cost of generating the initial complete plan (which is likely more expensive than generating a single reasoning step). Additionally, RAP uses 4-shot prompting while EvoPlan uses zero-shot, which affects per-sample cost. Can the authors provide: (a) a breakdown showing whether initial plan generation is included in the reported timings, and (b) a controlled comparison using the same prompting strategy? The paper's main efficiency claim depends on clarifying this.
2. Why limit evaluation to mathematical reasoning? The framework appears general, so demonstrating it on at least one other domain (e.g., code generation, strategic planning, embodied tasks) would strengthen claims about broad applicability and help understand where the approach works well vs poorly.
3. Can the authors provide failure analysis of the critic agents? Specifically, what percentage of high-scoring plans (fitness > 0.85) produce incorrect final answers? How does critic accuracy correlate with final task performance?
4. How sensitive is performance to the critic prompts? Was prompt engineering performed to optimize them? If so, should that engineering effort be factored into the method's cost, especially when comparing to methods that use off-the-shelf prompting?

---

### Official Review · Reviewer_ufBU · 2025-11-01

**Soundness:** 3
**Presentation:** 4
**Contribution:** 3
**Rating:** 6
**Confidence:** 3

**Summary:**

This paper introduces EvoPlan, an agent-based framework that replaces costly execution-based evaluations in LLM planning with an efficient agentic fitness function. The method uses two specialized critic agents: one focused on logical consistency and the other on feasibility, which evaluate complete plans without requiring execution. EvoPlan integrates these agents within a tree-based evolutionary search, enabling global optimization of plans at a fraction of the computational cost of prior MCTS or ToT approaches. Experiments on mathematical and reasoning benchmarks show large gains in both accuracy and efficiency, along with an finding that plan refinement and execution rely on separable reasoning skills.

**Strengths:**

- EvoPlan achieves high accuracy while reducing GPU time by up to 90% compared to execution-based MCTS, addressing a cost bottleneck in LLM reasoning.

- Framing plan generation as a global evolutionary optimization problem with static, agentic evaluation is an original and well-motivated idea that extends the limits of planning methods.

**Weaknesses:**

- Missing ablations on search parameters: The paper does not include ablation studies on the parameters of the MCTS component (e.g., exploration constant \(c_{\text{exp}}\), population size, search depth) or the weights \(w_{LC}\) and \(w_F\) in the aggregate fitness score \(R(C)\). Such analysis would clarify how sensitive EvoPlan’s performance is to these design choices.

- Unverified accuracy of the agentic critics: The paper claims that “The Agentic Fitness Signal is Effective and Non-Redundant,” and supports this with Table 4 by removing either critic to show accuracy drops. However, this only measures the influence of the critics on end performance, not the validity of their signals. Since LLM outputs can vary probabilistically from run to run, it remains unclear whether the Logical Consistency and Feasibility agents produce scores that accurately reflect the true quality of individual plans. A stronger test would be to compare the agents’ scores against real execution outcomes for selected nodes, checking whether higher predicted scores correspond to better-executed plans. This would confirm that the critics provide accurate and reliable signals, rather than performance gains emerging from stochastic behavior.

**Questions:**

1. How sensitive is EvoPlan to the choice of \(c_{\text{exp}}\), search depth, or the weighting of critic signals \(w_{LC}, w_F\)? Have the authors tested whether the results remain stable under different configurations?

2. Could the authors provide a validation study comparing critic scores to actual execution-based outcomes for a subset of plans? This would help verify that Logical Consistency and Feasibility scores genuinely reflect plan quality rather than random variation.

3. Are there cases where the two critics give conflicting signals, and if so, how does EvoPlan align them during the evolutionary update?

---

### Official Review · Reviewer_jgpn · 2025-11-01

**Soundness:** 1
**Presentation:** 2
**Contribution:** 2
**Rating:** 2
**Confidence:** 4

**Summary:**

The paper introduces EvoPlan, an agent-based reasoning framework that separates planning and execution. It performs evolutionary search over plans using critic agents for logical consistency and feasibility, then executes the best plan once. Experiments on reasoning benchmarks show improved performance compared to prior methods.

**Strengths:**

- The framework is conceptually clear and the methodology is clearly described

- The paper is generally well-written and easy to follow.

- Code is provided for review.

**Weaknesses:**

- The proposed plan–then–execute paradigm raises concerns regarding its effectiveness. In our experience, prompting a model to first produce a plan and then execute it often yields worse results than directly applying CoT prompting. This limitation stems from the fact that the model must generate a plan based solely on the query, without access to any intermediate reasoning results, which may hinder its ability to form accurate or adaptive strategies.

- The evaluation relies on prompting the same model to generate both the Logical Consistency and Feasibility scores. In our experience and according to prior studies [1,2], self-evaluation is generally unreliable for reasoning tasks, as models tend to be affected by self-bias.

- Line 125: It is unclear how the model is guaranteed to produce improved plans when prompted with its own previous plans and self-generated feedback. Additional analysis or ablation studies are needed to verify that this iterative prompting process consistently leads to better plans.

- Table 2: The reported performance is comparable to existing methods (e.g., Qwen2.5-7B-Instruct with MR or ToT) on simple tasks. However, in Table 6, the proposed method achieves substantially higher accuracy on more challenging datasets (AIME-24, AIME-25), which is difficult to interpret.
   - As noted earlier, prompting a model to first generate a plan and then execute it typically yields worse performance than direct CoT prompting, especially on hard problems.
   - Both the proposed method and the baselines rely on self-evaluation, which tends to exacerbate errors on such tasks.
   -  The proposed method performs tree search only over plans, followed by a single execution, while several baselines conduct search over executions as well.

- The novelty of the approach appears limited. Encouraging a model to iteratively generate and refine plans via self-evaluation closely resembles prior works [3,4].

## Reference:
[1] Huang, Jie, et al. "Large Language Models Cannot Self-Correct Reasoning Yet." The Twelfth International Conference on Learning Representations.

[2] Jiang, Dongwei, et al. "Self-[in] correct: Llms struggle with discriminating self-generated responses." Proceedings of the AAAI Conference on Artificial Intelligence. Vol. 39. No. 23. 2025.

[3] Madaan, Aman, et al. "Self-refine: Iterative refinement with self-feedback." Advances in Neural Information Processing Systems 36 (2023): 46534-46594.

[4] Wang, Lei, et al. "Plan-and-solve prompting: Improving zero-shot chain-of-thought reasoning by large language models." arXiv preprint arXiv:2305.04091 (2023).

**Questions:**

Which base model is used for Table 7? What are the results when all three components (planning, evaluation, and execution) use the 1.5 B model?

---

### Meta-Review · Area_Chair_a7tU · 2025-12-30

**Summary:**

The main concern from most of the reviewers is that the method’s evaluation signal comes from LLM-prompted critics without convincing validation that the critic scores correlate with actual execution correctness, which means that the accuracy of the agentic critics is not verified. Reviewers also note limited novelty relative to prior self-refinement works, plus limited evaluation scope largely confined to math reasoning. No rebuttals are given. Thus, I recommend rejection.

**Reviewer Concerns:**

No rebuttals are provided. Thus, the reviewers' concerns won't be addressed.

**Reviewer Scores:**

No rebuttals are provided. Thus, the reviewers' scores will possibly remain the same.

---

### Decision · Program_Chairs · 2026-01-26

Reject